# CLASS-GROUPED-NORMALIZED-MOMENTUM AND FASTER HYPERPARAMETER EXPLORATION TO TACKLE CLASS IMBALANCE IN FEDERATED LEARNING

## ABSTRACT

Local class imbalance rooted at global imbalance poses a critical challenge in federated learning (FL), where underrepresented classes suffer from poor predictive performance yet cannot be addressed by standard centralized techniques due to privacy and heterogeneity constraints. We propose FedCGNM (Federated Class-Grouped Normalized Momentum), a client-side optimizer in FL that partitions classes into a small number of groups, maintains a momentum per group, normalizes each group momentum to unit length, and uses the summation of the normalized group momentums as an update direction. This design both equalizes gradient magnitude across majority and minority groups and mitigates the noise inherent in rare-class gradients. Additionally, a resampling mechanism is employed to further mitigate class imbalance. To select sampling rates at clients efficiently in small-client federations, we propose FedHOO, an X-armed-bandit (XAB) based algorithm that exploits federated parallelism that evaluates many combinations of two candidate rates per client at linear cost. Empirical evaluation on four public long-tailed benchmarks and a proprietary chip-defect dataset demonstrates that FedCGNM consistently outperforms baselines and that coupling with FedHOO yields further improvements in small-scale federation.

## 1 INTRODUCTION

Class imbalance remains a major challenge in federated learning (FL). When the global label distribution aggregated over all clients is long-tailed, minority classes are underrepresented in training, which degrades their predictive accuracy. We study this imbalance setting in FL, defined as the situation where the aggregated class proportions remain highly skewed regardless of local distributions. Prior works in centralized learning mitigate imbalance through loss reweighting, advanced sampling, or generative augmentation, but these techniques are difficult to deploy in FL because privacy constraints prevent data exchange and synthetic generators are either infeasible or produce unrealistic samples for sparse regimes or domain-specific tasks such as defect detection. For instance, when working with a chip-defect dataset, one of our primary evaluation tasks in this paper, synthetic defect images fail to capture the true geometric details of actual defects.

Another strand of work, Per-Class Normalization (PCN), normalizes each class-specific gradient to unit length, preventing any class from dominating the update (Francazi et al., 2023). PCN effectively decreases the loss for all classes and, because it operates directly on gradients, integrates readily with other techniques. However, PCN introduces two limitations: it is negatively affected by directional noise (since minority class gradients often misalign with the true descent direction), and the sum of many unit vectors can produce a scaling mismatch that destabilizes convergence when there are many classes. PCN does not work well even for a moderate number of classes.

To address these issues, we propose Federated Class-Grouped Normalized Momentum (FedCGNM). Instead of normalizing all $C$ class-specific gradients, we partition classes into a few groups (typically majority vs. minority) and maintain a momentum vector for each group. At each local step, the group momentum is updated and normalized to unit length, and a client computes its final update direction by summing the resulting unit vectors. By reducing the number of normalized vectors from $C$ to just a few, FedCGNM mitigates the scaling-mismatch problem while still giving every group equal

magnitude. Furthermore, incorporating momentum attenuates the directional noise from minority classes. An additional benefit concerns client alignment, a primary cause of performance degradation in FL (Dandi et al., 2022). Heterogeneous clients often produce gradients whose magnitudes per class differ widely, amplifying misalignment. Because FedCGNM forces each group update to have the same unit norm, the aggregated directions across clients become more aligned.

Furthermore, during local training, each client applies resampling to reduce directional noise, so selecting the appropriate sampling rates is crucial. To find sampling rates efficiently, particularly for small-client settings, we introduce FedHOO, an algorithm based on the X-armed bandit framework that exploits the inherent parallelism in FL. In every communication round, FedHOO requests each client to train with only two candidate rates, yet by linearly combining the returned updates the server can infer the validation performance metric for all $2^K$ rate combinations, where $K$ is the number of clients. The method therefore identifies effective sampling rates, while avoiding an exhaustive sweep of the hyperparameter space corresponding to sampling rates.

Across four public benchmarks, our methods consistently outperforms traditional reweighting and sampling baselines, with gains up to 29% over FedAvg combined with resampling. In a large-scale industrial chip defect dataset, our method achieves a 16% improvement over the best baseline. Our main contributions are as follows.

1. **FedCGNM optimizer.** We introduce *FedCGNM*, the first client-side optimizer that groups labels, applies unit-norm momentum per group to balance majority and minority influence while reducing noise and scale mismatch. We also prove convergence matching the best-known FL rates under standard smoothness and variance assumptions.

2. **Variance-aware grouping rule.** We frame class partitioning as minimization of within-group variance and design a hyperparameter-free, data-driven threshold search on the class-frequency histogram. This is the first class grouping algorithm targeting class imbalance in FL. Empirically, this rule produces the optimal split found by exhaustive search, yielding a principled yet lightweight grouping strategy.

3. **FedHOO sampling-rate tuner.** While a grid search for sampling rates is common, more refined strategies are advantageous when the number of clients is low. The proposed X-armed-bandit-based exploration scheme is the first optimization based algorithm to determine combinatorial local sampling rates in FL. The method performs rapid, privacy-preserving search trading off exploration and exploitation.

4. **Strong performance** We validate our framework across multiple public benchmarks and a real-world industrial semiconductor chip-defect dataset. Across all settings, it consistently shows improvement over strong baselines.

## 2 RELATED WORKS

Class imbalance poses a significant challenge in supervised learning, where limited data from minority classes leads to biased models and poor performance on those classes (Chen et al., 2024b; Johnson & Khoshgoftaar, 2019). Common training-level solutions include re-weighting, which adjusts learning based on class frequency, and resampling, which alters the class distribution in training data.

**Reweighting Methods**   Most reweighting methods adjust each sample's contribution within the loss function to counteract class imbalance. In particular, weighted cross-entropy (Aurelio et al., 2019) assigns higher loss weights to minority-class samples, and focal loss (Lin et al., 2017) further down-weights well-classified majority instances to concentrate learning on harder, underrepresented cases. Class-balanced loss (Cui et al., 2019) computes weights based on the effective number of samples per class, thereby reflecting diminishing returns of additional samples. Beyond loss-level re-weighting, PCN (Francazi et al., 2023) rescales each class-specific gradient to unit norm, equalizing per-class influence during optimization. He (2024) introduces a technique to adjust the weight of gradient dynamically in a class-incremental learning scenario. They consider reweighting in class-level which works poorly as the number of classes increases, and only consider balance of the gradient magnitude. Different from them, we focus on solving the scaling issue and the directional noise problem (in addition to learning in the FL setting).

Several methods have tailored re-weighting to FL setting. FedGR (Guo et al., 2023) introduces an imbalanced softmax function accompanied by a gravitational regularizer to pull minority-class representations toward balanced decision boundaries. FL Ratio Loss (Wang et al., 2021) builds on centralized Ratio Loss by estimating global class proportions via secure aggregation and adjusting local losses accordingly. FedNoRo (Wu et al., 2023) leverages knowledge distillation and distance-aware aggregation to align client models, and incorporates a logical adjustment mechanism to address both data heterogeneity and class imbalance. Unlike these methods that focus on loss functions or models, we tackle class imbalance at the gradient level, pairing with a simple resampling technique.

**Resampling and Data Synthesis** Traditional resampling methods balance class balance by removing majority samples (under-sampling) or replicating minority ones (over-sampling) (Carvalho et al., 2025). More recent techniques like SMOTE (Chawla et al., 2002), GAMO (Mullick et al., 2019), and I-GAN (Pan et al., 2024) generate synthetic minority data and show strong empirical performance. However, when minority samples are extremely scarce (Chen et al., 2024a) or synthetic data risks being unrealistic or mislabeled (Alkhawaldeh et al., 2023), such methods become less viable. Thus, the paper confines itself to conventional under- and over-sampling without synthetic generation since we focus on those situations.

Choosing how much to re-sample remains an open problem: systematic investigations in centralized deep learning reveal that the optimal under- or over-sampling rate depends jointly on the dataset size and the severity of class skew (Buda et al., 2018). Curriculum-based schemes, such as Dynamic Curriculum Learning (Wang et al., 2019), further highlight the need to adapt sampling ratios over the course of training rather than fixing them a priori. In the federated setting, Düsing et al. (2024) cast client-side resampling as a tunable policy, optimized to minimize the global loss while respecting privacy constraints. FAST (Wang et al., 2023) advances this idea by viewing each sampling ratio as an arm in a multi-armed-bandit framework, enabling dynamic exploration during training. However, it treats local sampling rates independently, overlooking the combinatorial nature of federated learning. Our method adopts this adaptive philosophy, particularly suited to small-client federations, while addressing the combinatorial optimization challenge.

## 3 METHODOLOGY

Consider a federated learning system with $K$ clients for a $C$-class classification task. We assume that $n^{(1)} \geq n^{(2)} \geq \cdots \geq n^{(C)}$ where $n^{(c)}$ is the number of samples in class $c$ and the classes are indexed as $\{1, 2, \ldots, C\}$. The global loss function $f : \mathbb{R}^n \to \mathbb{R}$ is $f(x) = \sum_{k=1}^K p_k f_k(x)$, where $f_k(x)$ is a local loss, and $p_k$ is the weight of client $k$.

The overall algorithm is outlined in Algorithm 1. We first adopt a resampling strategy to adjust the client-side data distribution, improving representation of minority classes. Any resampling scheme that determines the local resampling rates can be used. After resampling, each client groups classes based to its effective label distributions. Local training is then performed using FedCGNM.

### 3.1 FEDERATED CLASS-GROUPED-NORMALIZED-MOMENTUM

In multi-class settings, Per-Class Normalization (PCN) rescales every class gradient to unit norm, equalizing magnitudes but leaving two major drawbacks. First, summing the resulting $C$ norm-one vectors produces an update whose norm varies between zero and $C$, creating severe scaling variability and impeding convergence. Second, normalization does not mitigate directional noise, and handling each class separately can overfit minority labels. Finally, when $C$ is large, many classes may be absent from a mini-batch, and computing/storing $C$ separate gradients becomes computationally and memory intensive.

The next-to-be-proposed FedCGNM addresses these issues by merging the classes into a small number $H \ll C$ of groups (typically majority and minority) and maintaining a momentum per group. At communication round $t$, the server broadcasts the global model $x^{(t)}$ and local sampling rates $r_k^{(t)}$ based on a sampling strategy (see Section 3.3), and each client constructs a partition $\{\mathcal{G}_{k,h}^{(t)}\}_{h=1}^H$ of the classes. Section 3.2 details the grouping rule. The local loss decomposes as $f_k(x) = \sum_{h=1}^H f_{k,h}^{(t)}(x)$, with $f_{k,h}^{(t)} := f_{k,h}(x^{(t)})$ being the sum of the loss functions of $x^{(t)}$ over the samples in $\mathcal{G}_{k,h}^{(t)}$.

---

**Algorithm 1** FedCGNM combined with Resampling Strategy

---

**Require:** global rounds $T$, local steps $E$, step size $\eta$, momentum factor $\beta$, client weights $\{p_k\}_{k=1}^K$
1: initialize global model $x^{(0)}$
2: **for** $t = 0, \ldots, T-1$ **do**
3:     server determines sampling rates $\{r_k^{(t)}\}_{k=1}^K$ based on resampling strategy (Sec. 3.3)
4:     server broadcasts $(x^{(t)}, r_k^{(t)})$ to every selected client $k \in \mathcal{K}^{(t)}$
5:     **for each** client $k \in \mathcal{K}^{(t)}$ **in parallel do**
6:         resample each class $c$ to have $\left(\frac{n^{(1)}}{n^{(c)}}\right)^{r_k^{(t)}}$ samples. Subsequent steps use the resampled data.
7:         construct groups $\{\mathcal{G}_{k,h}^{(t)}\}_{h=1}^H$ via the Grouping rule (Sec. 3.2)
8:         $x_k^{(t,0)} \leftarrow x^{(t)}; \quad m_{k,h}^{(t,0)} \leftarrow 0, \forall h$
9:         **for** $i = 1$ to $E$ **do**
10:            compute $g_{k,h}^{(t,i)}$
11:            $m_{k,h}^{(t,i)} \leftarrow \beta m_{k,h}^{(t,i-1)} + (1-\beta) g_{k,h}^{(t,i)}, \forall h$
12:            $x_k^{(t,i)} \leftarrow x_k^{(t,i-1)} - \eta \sum_{h=1}^{H} \frac{m_{k,h}^{(t,i)}}{\|m_{k,h}^{(t,i)}\|}$
13:        **end for**
14:        upload $x_k^{(t,E)}$ to server
15:    **end for**
16:    $x^{(t+1)} \leftarrow \sum_{k \in \mathcal{K}^{(t)}} p_k x_k^{(t,E)}$
17: **end for**
18: **return** final model $x^{(T)}$

---

During local training, client $k$ updates, for each group $h$ and step $i$, the momentum

$$m_{k,h}^{(t,i)} = \beta m_{k,h}^{(t,i-1)} + (1-\beta) g_{k,h}^{(t,i)}, \quad m_{k,h}^{(t,0)} = 0, \tag{1}$$

where $g_{k,h}^{(t,i)}(x) = \nabla f_{k,h}^{(t)}(x; \xi_k^{(t,i)})$ is the stochastic gradient computed on samples in mini-batch $\xi_k^{(t,i)}$ drawn solely from group $\mathcal{G}_{k,h}^{(t)}$, and $\beta \in [0,1)$ is the momentum factor. The per-step update direction is obtained by normalizing each momentum and summing across the $H$ groups as $x_k^{(t,i)} = x_k^{(t,i-1)} - \eta \sum_{h=1}^{H} \frac{m_{k,h}^{(t,i)}}{\|m_{k,h}^{(t,i)}\|}$ with learning rate $\eta > 0$. After $E$ iterations the client returns $x_k^{(t,E)}$ to the server, which aggregates by $x^{(t+1)} = \sum_{k \in \mathcal{K}^{(t)}} p_k x_k^{(t,E)}$ where $\mathcal{K}^{(t)}$ is the set of active clients at round $t$. Operating on a handful of group momenta stabilizes the step norm, suppresses directional noise, and remains computationally efficient even when the number of classes is large.

### 3.2 GROUPING OF CLASSES

We formulate the problem of partitioning $\{1, \ldots, C\}$ into $H$ disjoint groups as one-dimensional variance reduction on the class proportions. Let $q_c$ be the (resampled) proportion of class $c$, with $\sum_{c=1}^C q_c = 1$, and we assume $q_1 \geq \cdots \geq q_C$. We treat $\{q_c\}_{c=1}^C$ as points on the real line and select $H-1$ thresholds to form contiguous groups in which proportions are as similar as possible.

For a partition $G = \{\mathcal{G}_h\}_{h=1}^H$, define the group mass $S_h = \sum_{c \in \mathcal{G}_h} q_c$, group mean mass $\mu_h = S_h/|\mathcal{G}_h|$, and the within-group distribution $w_{c|h} = q_c/S_h$. In a mini-batch of size $B$, let $N_h \sim$ Binomial$(B, S_h)$ be the sample counts of group $h$ and $N_{h,c}$ be the sample counts of class $c$ in group $h$. Define the empirical share vector $\hat{\boldsymbol{w}}_h = (\hat{w}_{c|h})_{c \in \mathcal{G}_h}$ with $\hat{w}_{c|h} = N_{h,c}/N_h$, and compare it with the uniform target $u_h = (1/|\mathcal{G}_h|, \ldots, 1/|\mathcal{G}_h|) \in \mathbb{R}^{|\mathcal{G}_h|}$. We define group imbalance vector $\Delta_h = \frac{N_h}{B}(\hat{\boldsymbol{w}}_h - u_h)$. Taking expectation, we obtain

$$\mathbb{E}\|\Delta_h\|^2 = \left(S_h^2 + \frac{S_h(1-S_h)}{B}\right)\|\boldsymbol{w}_h - u_h\|^2 + \frac{S_h}{B}\left(1 - \sum_{c \in \mathcal{G}_h} w_{c|h}^2\right),$$

so the dominant term is $S_h^2\|\boldsymbol{w}_h - u_h\|^2$ with $O(1/B)$ corrections.

We next link this imbalance to the variance of raw proportions. Using $\|\boldsymbol{w}_h - u_h\|^2 = \sum_{c \in \mathcal{G}_h} \left( \frac{q_c}{S_h} - \frac{1}{|\mathcal{G}_h|} \right)^2 = \frac{|\mathcal{G}_h|}{S_h^2} \sigma_h^2$ where $\sigma_h^2 = \frac{1}{|\mathcal{G}_h|} \sum_{c \in \mathcal{G}_h} (q_c - \mu_h)^2$, the dominant term becomes $S_h^2\|\boldsymbol{w}_h - u_h\|^2 = |\mathcal{G}_h| \sigma_h^2$. Summing over groups and normalizing per class yields $\frac{1}{C} \sum_{h=1}^{H} |\mathcal{G}_h| \sigma_h^2 = \sum_{h=1}^{H} \omega_h \sigma_h^2$ with $\omega_h = |\mathcal{G}_h|/C$, which is the within-group variance objective on the sorted $\{q_c\}_{c=1}^{C}$. Consequently, selecting $H - 1$ thresholds by this strategy produces the partition that minimizes the class-balanced within-group dispersion and, by the argument above, asymptotically minimizes expected per-batch imbalance.

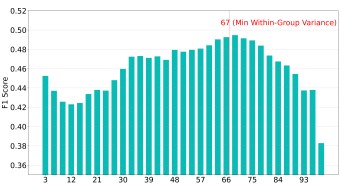

Figure 1: Test accuracy on CIFAR-100-LT ($\xi = 20, K = 5$) w.r.t. the number of classes assigned to the minority group. The red line marks the threshold selected by our grouping rule.

Figure 1 reports the test accuracy obtained when we exhaustively vary the split point between majority and minority classes. A threshold $t$ assigns the $t$ rarest classes to the minority group and the remaining $C - t$ classes to the majority group. Accuracy rises to a clear maximum near $t = 69$ and declines when either too few or too many classes are treated as minority. The red line marks the threshold chosen by our variance-based grouping rule, which coincides with the empirical optimum. Results of additional experiments in Appendix D.1 exhibit the same pattern, confirming that minimizing within-group variance serves as a reliable proxy for an exhaustive threshold search.

In summary, variance-aware grouping via minimizing the within-group variance on empirical class distribution provides a data-driven, lightweight mechanism that, when coupled with normalized momentum, substantially attenuates gradient noise while ensuring balanced directional contributions from majority and minority classes.

### 3.3 RESAMPLING

We employ a resampling strategy to further mitigate directional noise and increase learning in minority classes. In particular, we define the sampling rate $r \geq 0$ such that, for the class distribution, we oversample the class $c$ so that it has $\left( \frac{n^{(1)}}{n^{(c)}} \right)^r$ samples. For example, if $r = 0$, we do not resample, and if $r = 1$, we oversample so that all classes have the same rates of appearance in a mini-batch.

A possible way to find sampling rates is to view the problem through the lens of the X-armed bandit (XAB) over the continuous domain $\mathcal{X} = [r_{\min}, r_{\max}]^K$. An arm $r = (r_1, \ldots, r_K) \in \mathcal{X}$ specifies client rates, and pulling $r$ consists of executing one FL round where client $k$ trains with $r_k$, aggregating the local models, and evaluating the aggregated model. The reward $f(r)$ is the performance of the resulting aggregated model, and the learner must balance exploration and exploitation to identify a near-optimal $r \in \mathcal{X}$. Hierarchical Optimistic Optimization (HOO, Bubeck et al. (2011)) addresses XAB by creating a hierarchical, tree-based partition of the search space $\mathcal{X}$. The algorithm iteratively navigates this tree, selecting the most promising subregions to explore based on an optimistic estimate of their potential reward.

**FedHOO** However, standard XAB methods (e.g., exhaustive search or HOO) converge too slowly under FL's limited rounds and early sensitivity to hyperparameters. We introduce FedHOO, which exploits the parallelism of FL and is suitable for a small number of clients. FedHOO retains HOO's tree of boxes and enables much faster exploration by letting clients perform local training with two rates and exploring all combinations of local rates for validations.

The space $\mathcal{X} = [r_{\min}, r_{\max}]^K$ is organized as a $2^K$-ary tree. The root covers the entire search space, and each child $\nu$ halves its parent's range, and the midpoint $c(\nu)$ of node $\nu$'s interval acts as its representative point. Each node $\nu$ in depth $h$ stores a box $I(\nu) = \{[L_k(\nu), U_k(\nu)]\}_{k=1}^{K}$, an evaluation count $N(\nu)$, a running reward estimate $V(\nu)$, and an optimistic score $B(\nu)$.

At round $t$, the server selects the leaf $\nu$ with largest $B(\nu)$. Let $\mathcal{I}(\nu) = \prod_k [L_k, U_k]$ be its box and $h$ be its depth in the tree. For each client $k$, the server sends two rates $r_k^L = (3L_k + U_k)/4$, $r_k^U = (L_k + 3U_k)/4$, and the current model $x^{(t)}$. Client $k$ trains two local models (one per rate) and returns the deltas. By linearly combining these deltas, the server aggregates the models corresponding to **all** $2^K$ lower/upper choices, thereby evaluating the rewards of the $2^K$ child nodes in one round.

The optimistic score of $\nu$ is

$$B(\nu) = V(\nu) + \tau \operatorname{diam}(\nu)^h + \sqrt{\tfrac{\alpha \ln(t+1)}{N(\nu)}}, \tag{2}$$

where $\operatorname{diam}(\nu)$ is the diameter of $\mathcal{I}(\nu)$ and $\tau, \alpha > 0$ are hyperparameters. The second term of $B(\nu)$ is borrowed from UCB bandits (Auer et al., 2002) exploration. In FedHOO, server also keep track of reward $V(\nu)$ which is updated as exponential moving average of validation accuracies. Finally, the server expands $\nu$ into its $2^K$ child nodes corresponding to all combinations of two rates per client, and sets $x^{(t+1)}$ to the one with the highest reward among the $2^K$ candidates just evaluated.

Doubling each client's local training time unlocks an exponential exploratory gain. Enumerating those combinations explicitly would be prohibitive, but FedHOO obtains the same information at linear cost by utilizing parallelism of FL, making it vastly more efficient than existing search methods.

The strategy is especially advantageous in small federations, a configuration frequently encountered in industrial deployments. We therefore apply FedHOO when the number of clients is low. When $K$ is large, we revert to a uniform global sampling rate because significant exploration cannot be completed within a reasonable training budget and a single rate promotes update alignment, a consideration that becomes increasingly critical as $K$ grows. The entire algorithm is presented in Algorithm 2.

### 3.4 CONVERGENCE ANALYSIS OF FEDCGNM

We analyze the convergence of FedCGNM in isolation from the resampling component. In Algorithm 2, the resampling rates and groupings change by iteration, making our analysis complicated. Throughout this section we therefore assume static rates and groupings, i.e., $r_k = r_k^{(t)}$, $\mathcal{G}_{k,h} = \mathcal{G}_{k,h}^{(t)}$, for every $t$. The resulting algorithm is still non-trivial to analyze because the update sums unit normalized momenta and the per-group direction is a nonlinear biased transform of the stochastic gradient, so $\mathbb{E}[m/\|m\|] \neq \nabla f/\|\nabla f\|$. For the theoretical convergence analysis, we make the following assumptions.

**Assumption 3.1** (Smoothness). *Each local loss function $f_k$ is $L$-smooth, that is, for all $x$ and $y$,*

$$\|\nabla f_k(x) - \nabla f_k(y)\| \leq L \|x - y\|. \tag{A.1}$$

**Assumption 3.2** (Uniform bound). *There exist $G, \delta > 0$ such that, for all $x$ and for any $k, h$, we have*

$$\mathbb{E}\|\nabla f_{k,h}(x; \xi_k)\| \leq G, \quad \mathbb{E}\|m_{k,h}\| \geq \delta. \tag{A.2}$$

**Assumption 3.3** (Unbiasedness and bounded variance). *There exists $\sigma^2 > 0$ such that for any $x$ and $k$, we have*

$$\mathbb{E}[\nabla f_k(x; \xi_k)] = \nabla f_k(x), \tag{A.3}$$

$$\mathbb{E}\|\nabla f_k(x; \xi_k) - \nabla f_k(x)\|^2 \leq \sigma^2. \tag{A.4}$$

**Assumption 3.4.** *There exist $\gamma \in (0, 1]$ and $\kappa > 0$ such that for all $x$ and for any $k, h$, either*

$$\frac{\langle \nabla f_{k,h}(x^{(t)}), m_{k,h} \rangle}{\|\nabla f_{k,h}(x^{(t)})\| \|m_{k,h}\|} \leq 1 - \gamma \quad or \quad \big| \|m_{k,h}\| - \|\nabla f_{k,h}(x^{(t)})\| \big| \geq \kappa. \tag{A.5}$$

Assumption 3.4 asserts that, at each client–class update, the stochastic gradient cannot be both perfectly aligned *and* equal in length to its exponentially averaged momentum. One of the two gaps is virtually guaranteed in realistic training, and the assumption is strictly weaker than the simultaneous angle-and-norm bounds adopted in earlier analyses of normalized momentum methods.

The following theorem shows that FedCGNM attains the standard $\mathcal{O}(T^{-1/2})$ stationary-point rate under smoothness and bounded-variance assumptions. A complete proof is given in Appendix A.

**Theorem 3.5.** *Let Assumptions 3.1–3.4 hold and let $\beta = 1 - c\eta$ with a constant $c > 0$ satisfying $c\eta < 1$. Let $\{x_t\}_{t=1}^{T+1}$ be the iterates produced by FedCGNM, and let each client perform $E \geq 1$ local steps. Define $\Delta_0 = \mathbb{E}[f(x^{(0)})] - f_\star$, with a finite lower bound $f_\star = \min_x f(x)$. Then we have*

$$\frac{1}{T} \sum_{t=1}^{T} \mathbb{E}\|\nabla f(x^{(t)})\|^2 \leq \frac{2\Delta_0}{\eta E T} + \frac{2D_1\eta}{E} + \frac{2D_2\,\eta^2}{E}, \tag{3}$$

*where $D_1 = 2c\rho\sigma^2 E^2 H^2$, $D_2 = \frac{2}{3}\rho E^2 H^4 L^2 + \frac{2\beta^2}{1-\beta}\rho E^2 H^4 L^2$. Choosing the step size $\eta = \mathcal{O}(T^{-1/2})$ yields*

$$\min_{t<T} \mathbb{E}[\|\nabla f(x^{(t)})\|^2] \leq \mathcal{O}(T^{-1/2}). \tag{4}$$

## 4 EXPERIMENTS

### 4.1 EXPERIMENT SETTINGS

Our experiments evaluate FedCGNM and FedHOO on five classification benchmarks: two long-tailed image collections (CIFAR-10-LT and CIFAR-100-LT (Krizhevsky & Hinton, 2009)), two tabular datasets (Adult Income (Becker & Kohavi, 1996) and UNSW-NB15 (Moustafa & Slay, 2015)), and a semiconductor chip-defect dataset. The public image datasets CIFAR-10-LT and CIFAR-100-LT are produced using the standard long-tail protocol (Liu et al., 2019). For an imbalance rate $\xi$, the number of training samples in class $c$ (sorted by frequency), if we let $N_{\max}$ and $N_{\min}$ denote the numbers of samples of the majority and minority class, is

$$N_c = N_{\max} \cdot \xi^{-c/(C-1)}, \tag{5}$$

where $C$ denotes the total number of classes. If a dataset has balanced classes, we randomly sample from each class $c$ to have $N_c$ samples to make an imbalanced dataset. Test sets remain unaltered to test the performance of algorithms to produce balanced performance. The Adult Income dataset consists of 48,842 tabular records labeled as either "$\leq$ \$50 k" or "> \$50 k" with the minority class representing 23.9% of the total (the original imbalance rate of about 3.17). To test on stronger imbalance, we subsample the minority class until the imbalance rate $\xi = N_{\max}/N_{\min}$ reaches a higher number. UNSW-NB15 provides 175,341 network-flow records spanning ten classes with a natural imbalance of roughly 434. For Adult Income and UNSW datasets, we preserve the same class imbalance in the test splits as in their corresponding training sets to mirror real-world conditions.

The semiconductor Chip-Defect-Detection (CDD) corpus consists of approximately 780,000 high-resolution (224 × 224 after pre-processing) images captured on select years after 2020 from five factories. The ratio of defect images is 1.7%, with seven defect categories observed across hundreds of product types. Additional heterogeneity distributions are presented in Figure 2 and Figure 5. We split seventy percent of the dataset to training, fifteen percent to validation, and the remaining fifteen percent to test. We report the company's weighted accuracy that balances defect recall and non-defect precision. Some exact counts are withheld in accordance with the non-disclosure agreement. For full details of the implementation, including the selection of hyperparameters, see Appendix C.

We compare FedCGNM against four alternatives: (1) FedAvg (McMahan et al., 2017) with standard SGD, (2) FedAvg using class-weighted cross-entropy (Aurelio et al., 2019), (3) FedAvg with Ratio Loss (Wang et al., 2021), and (4) FedCGN, which adapts Per-Class Normalization to a grouped setting and employs CGN for local optimization. All methods share identical backbones and hyper-parameter schedules, but see Appendix for details of models, initial learning rate, batch size, weight decay and learning-rate scheduler. We use $H = 2$ for all grouping strategy implementations.

Two federation regimes are considered. In the small-scale scenario includes five clients with full participation. In the moderate-scale scenario, twenty clients are available but only half are sampled each round. Public datasets are partitioned IID across clients, whereas the chip corpus is divided according to natural factory boundaries, resulting in highly non-IID client data.

### 4.2 PRIMARY REAL-WORLD EVALUATION: CHIP-DEFECT DETECTION (CDD)

The semiconductor CDD task provides the realistic evaluation in our study since this dataset reflects actual production condition, with only 1.7% of samples exhibiting defects. Figure 3 shows that the

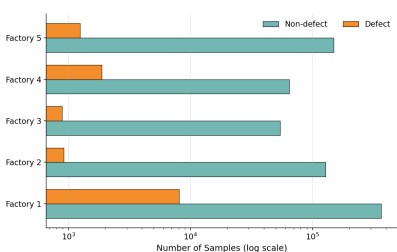

Figure 2: Number of defect and non-defect samples across five factories for CDD dataset.

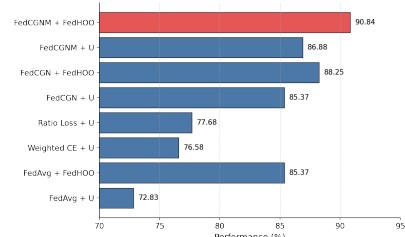

Figure 3: Test performance on the chip-defect dataset, with the metric scaled from 0 to 100.

Table 1: F1 scores on public benchmarks under two federation regimes. "C10" and "C100" abbreviate CIFAR-10 and CIFAR-100, and the trailing numbers denote the imbalance rates. The letter following each baseline denotes the resampling method, with "U" indicating a uniform global sampling rate. Gray numbers represent the standard deviation across three independent runs.

| Algorithm | C10-20 | C10-100 | C100-20 | C100-100 | Adult-3.17 | Adult-10 | Adult-20 | UNSW-434 |
|---|---|---|---|---|---|---|---|---|
| **K = 5** | | | | | | | | |
| FedAvg + U | 0.8259 ±0.002 | 0.6177 ±0.003 | 0.4458 ±0.002 | 0.2466 ±0.004 | 0.8432 ±0.001 | 0.9037 ±0.001 | 0.9443 ±0.003 | 0.7702 ±0.001 |
| Weighted CE + U | 0.8186 ±0.006 | 0.5379 ±0.007 | 0.3610 ±0.006 | 0.2183 ±0.011 | 0.8399 ±0.003 | 0.9059 ±0.002 | 0.9208 ±0.005 | 0.7210 ±0.013 |
| Ratio Loss + U | 0.8274 ±0.007 | 0.6183 ±0.002 | 0.4402 ±0.004 | 0.2734 ±0.007 | 0.8436 ±0.003 | 0.9068 ±0.003 | 0.9411 ±0.004 | 0.7740 ±0.008 |
| FedCGN + U | 0.8335 ±0.003 | 0.7054 ±0.005 | 0.4925 ±0.004 | 0.3116 ±0.005 | 0.8416 ±0.002 | 0.9106 ±0.001 | 0.9405 ±0.002 | 0.7778 ±0.004 |
| FedCGNM + U | 0.8568 ±0.004 | 0.7432 ±0.002 | 0.4983 ±0.003 | 0.3165 ±0.003 | 0.8456 ±0.002 | 0.9139 ±0.002 | 0.9458 ±0.001 | 0.7808 ±0.002 |
| FedCGNM + FedHOO | **0.8628** ±0.003 | **0.7485** ±0.004 | **0.5021** ±0.004 | **0.3183** ±0.006 | **0.8462** ±0.001 | **0.9148** ±0.003 | **0.9468** ±0.003 | **0.7823** ±0.001 |
| **K = 20** | | | | | | | | |
| FedAvg + U | 0.7128 ±0.004 | 0.4697 ±0.003 | 0.3544 ±0.002 | 0.1932 ±0.003 | 0.8452 ±0.001 | 0.9128 ±0.001 | 0.9424 ±0.001 | 0.7468 ±0.003 |
| Weighted CE + U | 0.6625 ±0.003 | 0.4426 ±0.004 | 0.3213 ±0.001 | 0.1807 ±0.002 | 0.8362 ±0.002 | 0.9050 ±0.002 | 0.9212 ±0.004 | 0.7120 ±0.005 |
| Ratio Loss + U | 0.7159 ±0.002 | 0.4563 ±0.004 | 0.3583 ±0.002 | 0.1950 ±0.003 | **0.8458** ±0.004 | 0.9131 ±0.002 | 0.9443 ±0.004 | 0.7414 ±0.006 |
| FedCGN + U | 0.7915 ±0.003 | 0.5210 ±0.004 | 0.3827 ±0.001 | 0.2138 ±0.003 | 0.8434 ±0.001 | 0.9101 ±0.003 | 0.9461 ±0.001 | 0.7733 ±0.003 |
| FedCGNM + U | **0.8010** ±0.002 | **0.5294** ±0.003 | **0.4051** ±0.002 | **0.2257** ±0.002 | 0.8453 ±0.001 | **0.9134** ±0.004 | **0.9474** ±0.002 | **0.7774** ±0.003 |

standard FedAvg baseline stalls at around 73, while loss reweighting and Ratio Loss provide only marginal improvements. FedCGN pushes performance into 85, and FedCGNM adds another improvement. Using FedHOO as resampling strategy consistently improves training. This demonstrates that variance-aware grouping, per-group momentum, and efficient rate exploration are not only effective on public benchmarks but also critical for real industrial deployments.

## 4.3 EVALUATION ON PUBLIC DATASETS

Table 1 summarizes F1 under two federation regimes. Across both federation regimes, FedCGNM outperforms all baselines by several points on average in all setting. In the small-client setting ($K = 5$), coupling FedCGNM with FedHOO yields the best accuracy in every scenario. Even without FedHOO, FedCGNM alone consistently beats competing optimizers. The only exception occurs under mild skew on Adult Income, where Ratio Loss briefly matches FedCGNM, suggesting that simple loss reweighting can suffice when imbalance is mild.

As the number of clients increases from five to twenty, accuracy declines for all methods due to greater heterogeneity. However, FedCGNM exhibits noticeably milder degradation compared to FedCGN, which is the best baseline: on average, FedCGN loses about 14–37% performance, while FedCGNM drops by only 7–32%. This indicates that group momentum not only boosts performance in small federations but also makes optimization more robust to client scaling.

## 4.4 SENSITIVITY ANALYSIS

**Sensitivity to Imbalance Severity**   On the multi-class image benchmarks (CIFAR-10 and CIFAR-100), all methods show declining F1 scores as the class distribution becomes more imbalanced, but FedCGNM's drop is noticeably milder. By contrast, on the Adult Income task, FedCGNM offers only a slight improvement, reflecting the relative ease of a binary prediction problem.

**Effect of Grouping and Number of Groups**   We examine the impact of the number of groups using training and validation losses for $H = 2, 3, 4$ in Figure 4 and Table 2. For additional results,

Table 2: F1 score for FedCGNM with different group counts $H$.

| Dataset | $K$ | $H = 2$ | $H = 3$ | $H = 4$ |
|---------|-----|---------|---------|---------|
| CF10-LT20 | 5 | 0.8565 | 0.8400 | 0.8341 |
|           | 20 | 0.7915 | 0.7916 | 0.7148 |
| CF10-LT100 | 5 | 0.7432 | 0.7328 | 0.7171 |
|            | 20 | 0.5210 | 0.5185 | 0.4957 |
| CF100-LT20 | 5 | 0.4983 | 0.4941 | 0.4485 |
|            | 20 | 0.4051 | 0.3916 | 0.3547 |
| CF100-LT100 | 5 | 0.3165 | 0.3121 | 0.2721 |
|             | 20 | 0.2257 | 0.2253 | 0.2007 |

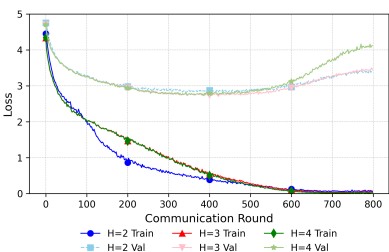

Figure 4: Training and validation loss on CIFAR-100-LT (imbalance rate = 20, $K = 20$) for FedCGNM with different group counts $H$.

see Appendix D.5). With two groups, validation loss remains lowest and most stable, whereas three or four groups lead to earlier increases in validation loss despite continued training loss reduction, indicating accelerated overfitting. On CIFAR-100, $H = 3$ occasionally yields slightly better accuracy, but the gap is marginal, so two groups offer the best trade-off between balanced class influence and generalization. Beyond the number of groups, our grouping strategy based on minimizing within-group variance provides clear benefits compared to naive half-splits: on CIFAR-10-LT20 with $K = 5$, our rule achieves an F1 score of 0.8565 versus 0.8382, and on CIFAR-100-LT20 with $K = 5$, it improves from 0.4339 to 0.4983. These gains confirm that variance-aware grouping better balances class contributions and yields superior generalization across datasets.

**Sensitivity to Group Momentum Factor**
We perform an ablation study on the group momentum factor $\beta$ across multiple datasets and client settings. The results in Table 3 show that very small ($\beta = 0$, which is Fed-CGN) or very large ($\beta = 0.9$) values generally underperform, while moderate values yield the strongest results. For example, on CIFAR-10-LT20 with $K = 5$, accuracy improves from 83.35 at $\beta = 0$ to a peak of 85.68 at $\beta = 0.5$, and on CIFAR-100-LT20 with $K = 20$, performance rises from 38.27

Table 3: Accuracy (%) across different $\beta$ values.

| Dataset | K | 0 | 0.1 | 0.3 | 0.5 | 0.7 | 0.9 |
|---------|---|---|-----|-----|-----|-----|-----|
| C10-20 | 5 | 83.35 | 83.72 | 84.21 | **85.68** | 84.27 | 83.93 |
|        | 20 | 79.15 | 79.55 | 79.55 | **80.10** | 79.50 | 79.54 |
| C10-100 | 5 | 70.54 | 71.11 | 71.11 | **74.32** | 71.11 | 71.11 |
|         | 20 | 52.10 | 52.34 | 52.35 | **52.94** | 52.65 | 52.68 |
| C100-20 | 5 | 49.25 | 49.33 | 49.35 | **49.83** | 49.46 | 49.76 |
|         | 20 | 38.27 | 39.16 | 39.52 | **40.51** | 39.84 | 40.24 |
| C100-100 | 5 | 31.16 | 31.22 | 31.08 | **31.65** | 30.95 | 30.03 |
|          | 20 | 21.38 | 21.92 | 22.22 | **22.57** | 22.56 | 22.48 |

to 40.51 at the same setting. Similar trends appear consistently across datasets, indicating that moderate momentum factors strike the right balance between stability and adaptivity, thereby offering the best overall performance.

Additional experiments (Appendix D) confirm that FedHOO accelerates sampling-rate search compared to standard HOO, and FedCGNM maintains its advantage under non-IID distributions, and large-client federations. These results demonstrate the robustness and generality of our approach across diverse and challenging federated learning conditions.

## 5 CONCLUSION

We have introduced FedCGNM, a simple yet effective client-side optimizer that balances majority and minority class influence by grouping labels and applying normalized momentum per group, and FedHOO, an efficient federated sampling rate exploration strategy that exploits parallelism of FL to explore sampling rates without exhaustive search. Our convergence analysis shows that FedCGNM matches the best known rates for momentum-based FL under standard assumptions, and empirical results on both public benchmarks and a chip-defect detection dataset demonstrate its consistent superiority over prior reweighting, sampling, and per-class normalization methods. Together, these components form a practical framework for mitigating global class imbalance in privacy-preserving federated settings.

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

# A  PROOF

## A.1  SUPPLEMENTARY LEMMAS

**Lemma A.1.** *Let Assumptions 3.2 and 3.4 hold. For every client $k$, group index $h$, global round $t$, and local step $i$,*

$$\left\| \nabla f_{k,h}(x^{(t)}) - \frac{m_{k,h}^{(t,i)}}{\|m_{k,h}^{(t,i)}\|} \right\|^2 \leq \rho \left\| \nabla f_{k,h}(x^{(t)}) - m_{k,h}^{(t,i)} \right\|^2, \qquad \rho := \max\left\{ \frac{G^2+1+2G}{\delta^2\gamma(2-\gamma)}, \frac{G^2+1+2G}{\kappa^2} \right\}.$$

$$\tag{6}$$

*Proof.* Fix $(k, h, t, i)$ and set $a := \nabla f_{k,h}(x^{(t)})$, $b := m_{k,h}^{(t,i)}$, $u := b/\|b\|$, $d := \|a\| \leq G$, $r := \|b\| \geq \delta$, $\theta := \angle(a, b)$. Define

$$R(d, r, \theta) := \frac{\|a - u\|^2}{\|a - b\|^2} = \frac{d^2 + 1 - 2d \cos \theta}{d^2 + r^2 - 2dr \cos \theta}.$$

Since $\|a - u\|^2 \leq d^2 + 1 + 2d \leq N_{\max} := G^2 + 1 + 2G$, it suffices to lower–bound the denominator in the two mutually exclusive cases of Assumption 3.4:

(i) *Angle gap* : $1 - \cos \theta \geq \gamma$ implies

$$\|a - b\|^2 = (r - d)^2 + 2dr(1 - \cos \theta) \geq (\delta - d)^2 + 2\gamma\delta d \geq \delta^2 \gamma(2 - \gamma).$$

(ii) *Norm gap* : $|r - d| \geq \kappa$ yields $\|a - b\|^2 \geq (r - d)^2 \geq \kappa^2$.

Hence $R(d, r, \theta) \leq \rho$ with $\rho$ given in equation 6; substituting $R$ completes the proof. $\qquad\square$

**Corollary A.2.** *Under Assumption 3.2 and 3.4, for every client $k$, class $h$, round $t$, and local step $i$,*

$$\mathbb{E}\left\|\nabla f_{k,h}(x^{(t)}) - \frac{m_{k,h}^{(t,i)}}{\|m_{k,h}^{(t,i)}\|}\right\|^2 \leq \rho\, \mathbb{E}\left\|\nabla f_{k,h}(x^{(t)}) - m_{k,h}^{(t,i)}\right\|^2, \qquad \forall k, h, t, i. \tag{7}$$

*Proof.* Because inequality equation 6 is valid pointwise, it holds for every outcome of the algorithm's randomness. Taking expectations on both sides preserves the inequality, yielding equation 7. $\qquad\square$

**Lemma A.3.** *Let $\{x_k^{(t,s)}\}_{s=0}^E$ and $m_{k,h}^{(t,i)}$ be the local iterate and momentum of client $k$ in communication round $t$, produced by FedCGNM. Under Assumption 3.1-3.3, for any group $h \in \{1, \ldots, H\}$, and local step $i \in \{1, \ldots, E\}$,*

$$\mathbb{E}[\|\nabla f_{k,h}(x_k^{(t,0)}) - m_{k,h}^{(t,i)}\|^2] \leq 2(i-1)^2 H^2 L^2 \eta^2 + \frac{2\beta^2}{1 - \beta} i H^2 L^2 \eta^2 + 2i(1 - \beta)^2 \sigma^2. \tag{8}$$

*Proof.* Throughout the proof the fixed indices $k, h, t$ are suppressed and the shorthands

$$g_s = \nabla f_{k,h}(x_k^{(t,s-1)}), \quad g_s^{\text{stoch}} = \nabla f_{k,h}(x_k^{(t,s-1)}; \xi_{k,h}^{(t,s)}), \quad m_s = m_{k,h}^{(t,s)}$$

are used, where $\xi_{k,h}^{(t,s)}$ denotes the mini-batch sampled at the $s$-th local step and $\mathbb{E}[g_s^{\text{stoch}}] = g_s$ with $\mathbb{E}\|g_s^{\text{stoch}} - g_s\|^2 \leq \sigma^2$ by Assumption 3.3. We note that, by assumption, the grouping remains fixed throughout training.

During each local step $s$ at round $t$, the model parameters of client $k$ are moved by

$$\eta\, d_k^{(t,s)} = \eta \sum_h \frac{m_{k,h}^{(t,s)}}{\|m_{k,h}^{(t,s)}\|},$$

where the update direction $d_k^{(t,s)}$ satisfies $\|d_k^{(t,s)}\| \leq H$. After $i - 1$ such steps the cumulative displacement is bounded by

$$\|x_k^{(t,i-1)} - x_k^{(t,0)}\| \leq (i - 1) H \eta.$$

By $L$-smoothness of $\nabla f_{k,h}$ this gives

$$\mathbb{E}\|g_1 - g_i\|^2 \leq H^2 L^2 (i - 1)^2 \eta^2. \tag{9}$$

For $s \geq 0$ define the difference $e_s := m_{k,h}^{(t,s)} - \nabla f_{k,h}(x_k^{(t,s-1)})$ and write $F_s := \mathbb{E}\|e_s\|^2$. Since we have, using $m_s = \beta m_{s-1} + (1 - \beta) g_s$,

$$e_s = \beta e_{s-1} + \beta(g_{s-1} - g_s) + (1 - \beta)(g_s^{\text{stoch}} - g_s),$$

expanding $\|e_s\|^2$ and taking full expectation gives

$$F_s = \beta^2 F_{s-1} + \beta^2 \mathbb{E}\|g_{s-1} - g_s\|^2 + (1-\beta)^2 \mathbb{E}\|g_s^{\text{stoch}} - g_s\|^2$$

$$+ 2\beta^2 \mathbb{E}\langle e_s,\, g_{s-1} - g_s \rangle \tag{I}$$

$$+ 2\beta(1-\beta)\mathbb{E}\langle e_{s-1},\, g_s^{\text{stoch}} - g_s \rangle \tag{II}$$

$$+ 2\beta(1-\beta)\mathbb{E}\langle g_{s-1} - g_s,\, g_s^{\text{stoch}} - g_s \rangle. \tag{III}$$

**Cross term (II).** Let $\mathcal{F}_s$ be the $\sigma$–algebra generated by all randomness up to step $s$. $e_{s-1}$ is measurable with respect to $\mathcal{F}_{s-1}$, whereas $g_s^{\text{stoch}} - g_s$ depends on the fresh mini-batch $\xi_{k,h}^{(t,s)}$. Hence $\mathbb{E}[\,g_s^{\text{stoch}} - g_s \mid \mathcal{F}_{-1}] = 0$ and

$$\mathbb{E}\langle e_{s-1},\, g_s^{\text{stoch}} - g_s \rangle = 0.$$

**Cross term (III).** The same independence argument yields $\mathbb{E}\langle g_{s-1} - g_s,\, g_s^{\text{stoch}} - g_s \rangle = 0$.

**Cross term (I).** For any $\alpha > 0$, Young's inequality gives

$$2\langle e_{s-1},\, g_{s-1} - g_s \rangle \le \alpha\|e_{s-1}\|^2 + \frac{1}{\alpha}\|g_{s-1} - g_s\|^2.$$

Multiplying by $\beta^2$ produces

$$2\beta^2 \mathbb{E}\langle e_{s-1},\, g_{s-1} - g_s \rangle \le \alpha\beta^2 F_{s-1} + \frac{\beta^2}{\alpha}\,\mathbb{E}\|g_{s-1} - g_s\|^2.$$

Choose $\alpha = \frac{1-\beta}{\beta} > 0$, so that $q := (1+\alpha)\beta^2 = \beta \in [0,1)$.

Combining all cross terms, and by using $\mathbb{E}\|g_{s-1} - g_s\|^2 \le H^2 L^2 \eta^2$ and $\mathbb{E}\|g_s^{\text{stoch}} - g_s\|^2 \le \sigma^2$, we have

$$F_s \le \beta\, F_{s-1} + \frac{\beta^2}{1-\beta}\, H^2 L^2 \eta^2 + (1-\beta)^2 \sigma^2. \tag{10}$$

Iterating equation 10 from $F_0 = 0$ gives

$$F_s \le (1-\beta^s)(1-\beta)\sigma^2 + \frac{\beta^2(1-\beta^s)}{(1-\beta)^2}\, H^2 L^2 \eta^2$$

$$\le (1-\beta)^2 s\sigma^2 + \frac{\beta^2}{1-\beta}\, sH^2 L^2 \eta^2.$$

Since $g_1 - m_i = (g_1 - g_i) + (g_i - m_i)$, $\|a + b\|^2 \le 2\|a\|^2 + 2\|b\|^2$ together with equation 9 yields

$$\mathbb{E}\|g_1 - m_i\|^2 \le 2H^2 L^2 (i-1)^2 \eta^2 + 2F_i,$$

which, upon substituting the bound on $F_i$ above, completes the proof of Lemma A.3. $\qquad\square$

## A.2 PROOF OF THEOREM 3.5

*Proof.* For each client $k$ and communication round $t$, the local update can be expressed as

$$x_k^{(t,E)} - x_k^{(t,0)} = -\eta \sum_{i=1}^{E} \sum_{h=1}^{H} \frac{m_{k,h}^{(t,i)}}{\|m_{k,h}^{(t,i)}\|} = -\eta E\, \Delta_k^{(t)}, \tag{11}$$

where we define $\Delta_k^{(t)} = \frac{1}{E}\sum_{i=1}^{E}\sum_{h=1}^{H}\frac{m_{k,h}^{(t,i)}}{\|m_{k,h}^{(t,i)}\|}$ as the average local update of client $k$ in the communication round $t$.

By $L$-smoothness assumption,

$$\mathbb{E}\left[f(x^{(t+1)})\right] \le \mathbb{E}\left[f(x^{(t)})\right] + \mathbb{E}\left[\langle \nabla f(x^{(t)}), x^{(t+1)} - x^{(t)} \rangle\right] + \frac{L}{2}\mathbb{E}\|x^{(t+1)} - x^{(t)}\|^2. \tag{12}$$

For the third term, by using Jensen's Inequality, we have

$$
\begin{aligned}
\mathbb{E}\|x^{(t+1)} - x^{(t)}\|^2 &= \mathbb{E}\left\|\sum_k p_k \cdot \eta \sum_{i=1}^{E}\sum_{h=1}^{H} \frac{m_{k,h}^{(t,i)}}{\|m_{k,h}^{(t,i)}\|}\right\|^2 \\
&\leq \eta^2 \sum_k p_k \mathbb{E}\left\|\sum_{i=1}^{E}\sum_{h=1}^{H} \frac{m_{k,h}^{(t,i)}}{\|m_{k,h}^{(t,i)}\|}\right\|^2 \\
&\leq \eta^2 \sum_k p_k E^2 H^2 \\
&\leq \eta^2 E^2 H^2.
\end{aligned}
\tag{13}
$$

By plugging (11) and (13) into (12), we have

$$
\mathbb{E}\left[f(x^{(t+1)})\right] - \mathbb{E}\left[f(x^{(t)})\right] \leq -\eta E\, \mathbb{E}\left[\langle \nabla f(x^{(t)}), \sum_k p_k \Delta_k^{(t)}\rangle\right] + \frac{\eta^2 E^2 H^2 L}{2}.
\tag{14}
$$

We apply $\langle a, b\rangle = \frac{\|a\|^2}{2} + \frac{\|b\|^2}{2} - \|a - b\|^2$, then we have

$$
\begin{aligned}
\mathbb{E}\left[f(x^{(t+1)})\right] - \mathbb{E}\left[f(x^{(t)})\right] &\leq -\eta E\left[\frac{1}{2}\mathbb{E}\|\nabla f(x^{(t)})\|^2 + \frac{1}{2}\mathbb{E}\left\|\sum_k p_k \Delta_k^{(t)}\right\|^2 - \mathbb{E}\left\|\nabla f(x^{(t)}) - \sum_k p_k \Delta_k^{(t)}\right\|^2\right] + \frac{\eta^2 E^2 H^2 L}{2} \\
&\leq -\frac{\eta E}{2}\mathbb{E}\|\nabla f(x^{(t)})\|^2 + \eta E\, \mathbb{E}\left\|\nabla f(x^{(t)}) - \sum_k p_k \Delta_k^{(t)}\right\|^2 + \frac{\eta^2 E^2 H^2 L}{2}.
\end{aligned}
\tag{15}
$$

Rewrite the discrepancy in equation 15 as

$$
\nabla f(x^{(t)}) - \sum_k p_k \Delta_k^{(t)} = \frac{1}{E}\sum_{k=1}^{K} p_k \sum_{i=1}^{E}\sum_{h=1}^{H}\left(\nabla f_{k,h}(x^{(t)}) - \frac{m_{k,h}^{(t,i)}}{\|m_{k,h}^{(t,i)}\|}\right)
\tag{16}
$$

Applying Cauchy–Schwarz and then Corollary A.2,

$$
\begin{aligned}
\mathbb{E}\left\|\nabla f(x^{(t)}) - \sum_k p_k \Delta_k^{(t)}\right\|^2 &\leq \frac{H}{E}\sum_k p_k \sum_{i=1}^{E}\sum_{h=1}^{H}\mathbb{E}\left\|\nabla f_{k,h}(x^{(t)}) - \frac{m_{k,h}^{(t,i)}}{\|m_{k,h}^{(t,i)}\|}\right\|^2 \\
&\leq \frac{\rho H}{E}\sum_k p_k \sum_{i=1}^{E}\sum_{h=1}^{H}\mathbb{E}\left\|\nabla f_{k,h}(x^{(t)}) - m_{k,h}^{(t,i)}\right\|^2.
\end{aligned}
\tag{17}
$$

Combining the descent bound equation 15, discrepancy bound equation 17, and the moment-difference bound of Lemma A.3 yields for every global round $t$

$$
\begin{aligned}
\mathbb{E}\left[f(x^{(t+1)})\right] - \mathbb{E}\left[f(x^{(t)})\right] &\leq -\frac{\eta E}{2}\mathbb{E}\|\nabla f(x^{(t)})\|^2 + \eta E\left(\frac{\rho H}{E}\sum_k p_k \sum_{i=1}^{E}\sum_{h=1}^{H}\mathbb{E}[\|\nabla f_{k,h}(x^{(t)}) - m_{k,h}^{(t,i)}\|^2]\right) + \frac{\eta^2 E^2 H^2 L}{2} \\
&\leq -\frac{\eta E}{2}\mathbb{E}\|\nabla f(x^{(t)})\|^2 + D_1\eta^2 + D_2\eta^3
\end{aligned}
\tag{18}
$$

where the deterministic coefficients are

$$D_1 := 2c\rho\sigma^2 E^2 H^2,$$

$$D_2 := \frac{2}{3}\rho E^2 H^4 L^2 + \frac{2\beta^2}{1-\beta}\rho E^2 H^4 L^2.$$

Re-arrainging equation 18 and summation over $T$ yields

$$\frac{1}{T}\sum_{t=1}^{T}\mathbb{E}\big\|\nabla f(x^{(t)})\big\|^2 \leq \frac{2(\mathbb{E}[f(x^{(1)})] - \mathbb{E}[f(x^{(T+1)})])}{\eta E T} + \frac{2D_1\eta}{E} + \frac{2D_2\eta^2}{E}. \tag{19}$$

Set $1 - \beta = c\eta$ for some $c > 0$ satisfying $c\eta < 1$. Then, for $\eta = \eta_0 T^{-1/2}$, we have

$$\frac{2(\mathbb{E}[f(x^{(1)})] - \mathbb{E}[f(x^{(T+1)})])}{\eta E T} = \frac{2(\mathbb{E}[f(x^{(1)})] - f^*)}{\eta_0 E \sqrt{T}} = \mathcal{O}(T^{-1/2})$$

$$\frac{2D_1\eta}{E} = 2c\rho\sigma^2 E^2 H^2 \eta = \mathcal{O}(T^{-1/2})$$

$$\frac{2D_2\eta^2}{E} = \left(\frac{4}{3}\eta + \frac{2\beta^2}{c}\right)\rho E H^4 L^2 \eta = \mathcal{O}(T^{-1/2}),$$

Therefore, the RHS of equation 19 is $\mathcal{O}(T^{-1/2})$. Taking the minimum over $t = 1, \ldots, T$ on the left and observing that each term is nonnegative gives the same upper bound for $\min_{t<T}\mathbb{E}[\|\nabla f(x^{(t)})\|^2]$. Hence,

$$\min_{t<T}\mathbb{E}[\|\nabla f(x^{(t)})\|^2] \leq \mathcal{O}(T^{-1/2}), \tag{20}$$

which completes the proof of Theorem 3.5. $\qquad\square$

# B   DETAIL OF FEDHOO

We present the pseudocode of FedHOO in Algorithm 2. We write $\mathrm{diam}(\nu) = \max_k \big(U_k(\nu) - L_k(\nu)\big)$ corresponding to the $\ell_\infty$ width of the interval and use $\odot$ as element-wise product.

**Intuition with an example.**   To illustrate how FedHOO works, consider a simple setting with $K = 2$ clients, and the search space is $[0, 1]^2$. The root node of the search tree corresponds to the full box $[0, 1] \times [0, 1]$. If we split this root into four quadrants, the child nodes are

$$[0, 0.5] \times [0, 0.5], \quad [0, 0.5] \times [0.5, 1.0], \quad [0.5, 1.0] \times [0, 0.5], \quad [0.5, 1.0] \times [0.5, 1.0].$$

Exploring further, the child $[0, 0.5] \times [0, 0.5]$ can itself be split into four sub-boxes such as $[0, 0.25] \times [0, 0.25]$ and $[0, 0.25] \times [0.25, 0.5]$, and so on. This recursive partitioning continues as the algorithm zooms in on promising regions.

In the standard HOO algorithm, only *one* node can be explored in each round. For example, at round 1, HOO would select just a single quadrant, say $[0, 0.5] \times [0.5, 1.0]$, and update its statistics based on a single evaluation. Over many rounds, this gradually builds information, but the search may proceed slowly because each evaluation provides feedback for only one region of the tree.

By contrast, FedHOO leverages the federated setting to explore many regions at once using parallelism of FL. Because each client performs two local runs (with two different probe rates), the server can synthesize outcomes corresponding to all corner combinations of the current interval. In the two-client example, this means that in a *single* round, FedHOO obtains rewards for all four quadrants simultaneously. For instance, at round 1, FedHOO evaluates the four children at depth $h = 1$:

$$[0, 0.5] \times [0, 0.5], \quad [0, 0.5] \times [0.5, 1.0], \quad [0.5, 1.0] \times [0, 0.5], \quad [0.5, 1.0] \times [0.5, 1.0].$$

---

**Algorithm 2** FedHOO

---

**Require:** bounds $r_{\min}, r_{\max}$, global rounds $T$, optimism constant $\alpha$
1: **initialize** $\nu_{\text{root}}$ with interval $\mathcal{I}(\nu_{\text{root}}) = [r_{\min}, r_{\max}]^K$; set $V(\nu_{\text{root}}) = 0$, $N(\nu_{\text{root}}) = 0$, $B(\nu_{\text{root}}) = +\infty$
2: **initialize** global model $x^{(0)}$
3: **for** $t = 0, \ldots, T - 1$ **do**
4:     $\nu \leftarrow \arg\max_{\text{leaf}} B(\nu)$; let $\mathcal{I}(\nu) = \prod_{k=1}^K [L_k, U_k]$
5:     **broadcast** $x^{(t)}$ and send $r_k^L = (3L_k + U_k)/4$, $r_k^U = (L_k + 3U_k)/4$ to each client $k$
6:     **for each** client $k$ **in parallel do**
7:         train with rate $r_k^L$ starting from $x^{(t)}$ to obtain local update $\Delta_k^L = x^{(t)} - \bar{x}^L$; $\bar{x}^L$ is the resulting solution of the local training with $r_k^L$
8:         train with rate $r_k^U$ starting from $x^{(t)}$ to obtain local update $\Delta_k^U = x^{(t)} - \bar{x}^U$; $\bar{x}^U$ is the resulting solution of the local training with $r_k^U$
9:         **return** $(\Delta_k^L, \Delta_k^U)$
10:    **end for**
11:    **for each** $s \in \{0, 1\}^K$ **do**
12:       Define $\hat{r}^{(s)} = (1 - s) \odot L + s \odot U$ and $\tilde{r}^{(s)} = s \odot L + (1 - s) \odot U$
13:       $\Delta^{(s)} = \sum_{k=1}^K p_k \big[ (1 - s_k)\Delta_k^L + s_k \Delta_k^U \big]$
14:       $x^{(s)} \leftarrow x^{(t)} + \Delta^{(s)}$,     $R^{(s)} \leftarrow \text{Validate}(x^{(s)})$
15:       create leaf $\nu_s$ with parent node $\nu$ with $I(\nu) = \Pi_k[\hat{r}_k^{(s)}, \tilde{r}_k^{(s)}]$
16:       set $V(\nu_s) \leftarrow R^{(s)}$, $N(\nu_s) \leftarrow 1$, $B(\nu_s) \leftarrow V(\nu_s)$
17:    **end for**
18:    $s^\star \leftarrow \arg\max_s R^{(s)}$,    $x^{(t+1)} \leftarrow x^{(s^\star)}$
19:    $\bar{R} \leftarrow 2^{-K} \sum_s R^{(s)}$
20:    **for node** $\nu'$ on path from $\nu$ to root **do**
21:       $N(\nu') \leftarrow N(\nu') + 2^K$
22:       $V(\nu') \leftarrow V(\nu') + \dfrac{2^K}{N(\nu')} \big( \bar{R} - V(\nu') \big)$
23:       $B(\nu') \leftarrow V(\nu') + \tau \, \text{diam}(\nu)^h + \sqrt{\alpha \ln(t + 1) / N(\nu')}$
24:    **end for**
25: **end for**

---

This parallel evaluation dramatically accelerates the search. Rather than spending four rounds to cover these quadrants (as in HOO), FedHOO requires only one. As the depth increases, the same principle applies: four sub-intervals at depth $h = 2$ can be evaluated together by reusing the two local runs per client.

**Summary and limitation.** The main advantage of FedHOO is that the federated setting allows the server to combine a small number of local runs into exponentially many synthetic evaluations. In the two-client example, two runs per client yield four corner evaluations in each round, and more generally $2^K$ corners can be evaluated from only two runs per client. This exponential coverage greatly accelerates the search compared to classical HOO, which can only evaluate a single node per round.

A limitation of this approach is the overhead of validating $2^K$ candidate models at each round, which may become expensive for large $K$. In addition, extending the procedure to partial participation is not straightforward, since missing client updates can prevent consistent synthesis of all corners. These issues suggest that while FedHOO is powerful for moderate $K$ or cluster-level dimensions, further work is needed to make it scalable to very large federations.

# C DETAIL OF THE EXPERIMENT SETTING

In this appendix, we provide full details of our experimental setup, including datasets, model architectures, training hyperparameters, and federated protocols.

## C.1 DATASETS AND MODELS

We evaluate on five benchmarks. CIFAR-10-LT and CIFAR-100-LT are long-tailed variants of CIFAR-10/100 with imbalance rates 20 and 100, constructed via the protocol of Liu et al. Liu et al. (2019). Adult Income is a binary classification task with original imbalance ratio 3.17 and additional settings of 10 and 20 obtained by subsampling the minority class Becker & Kohavi (1996). UNSW-NB15 is a ten-class network-flow dataset with natural imbalance 434 Moustafa & Slay (2015). The proprietary Chip-Defect-Detection (CDD) corpus comprises of approximately 780k optical micrographs (224×224) from five semiconductor fabs, in which only 1.7% of samples contain defects. CDD dataset is split by factory into 70% train, 15% validation, and 15% test. To expose the class imbalance inherent in our dataset, Figure 5 reports the number of defect and non-defect samples recorded by each factory.

We train a ResNet18 (He et al., 2016) model from scratch on all public image benchmarks using batch normalization for FedAvg, Ratio Loss, and Weighted Cross Entropy and using group normalization for FedCGN and FedCGNM. For tabular tasks we employ a four layer fully connected network with hidden dimensions of 32, 16, 8, and 2 and ReLU activations on the Adult Income data and we adopt the CNN–LSTM architecture of Pear & Kibria (2024), consisting of stacked one-dimensional convolutional filters feeding into an LSTM to capture temporal correlations in network-flow features for UNSW-NB15. For CDD, we use ResNet-34. Table 4 summarizes each architecture.

Table 4: Model specifications for each dataset used in our experiments.

| Dataset | Input | Backbone | Principal Layers |
|---|---|---|---|
| CDD | 224×224 RGB | ResNet-34 | conv $7 \times 7$–BN–ReLU; 4× residual stages |
| CIFAR-10-LT | 32×32 RGB | ResNet-18 | conv $3 \times 3$; 8× basic blocks |
| CIFAR-100-LT | 32×32 RGB | ResNet-18 | conv $3 \times 3$; 8× basic blocks |
| Adult Income | 104-dim tabular | 4-layer FFNN | $32{\rightarrow}16{\rightarrow}8{\rightarrow}2$ with ReLU and BN/GN |
| UNSW-NB15 | 196-length sequence | CNN–LSTM | Conv1D[128,256,512] $\rightarrow$ LSTM $\rightarrow$ FC |

## C.2 IMPLEMENTATION AND TRAINING DETAILS

All algorithms—FedAvg (SGD), weighted cross-entropy, Ratio Loss, FedCGN, and Fed-CGNM—share the same training search strategy. We tune the initial learning rate over the set $\{0.4, 0.2, 0.1, 0.05, 0.01\}$ by grid search and then decay it with a cosine annealing curve that reaches $10^{-4}$ at the final round. Weight decay is selected from $\{10^{-4}, 10^{-3}\}$ and the batch size is selected from $\{32, 64, 128\}$. For FedCGNM we additionally test the group momentum coefficient $\beta$ over $\{0.5, 0.6, 0.7, 0.8, 0.9\}$. Each communication round runs three local epochs when five clients participate and five local epochs when twenty clients participate on the image benchmarks; Adult Income and UNSW NB15 use a single local epoch per round because their training sets are comparatively small. All jobs are executed with PyTorch 2.4.1 and CUDA 11.4 on four NVIDIA TITAN Xp GPUs.

Sampling rates $r_k \in [0.4, 0.8]$ are tuned via FedHOO when $K = 5$, initializing a hierarchical tree over $[0.4, 0.8]^K$ with optimism constant $\alpha = 1.0$. In each round, clients evaluate two candidate rates $(r_k^L, r_k^U)$, return update deltas, and the server extrapolates rewards for all $2^K$ combinations at linear cost. For $K = 20$, a uniform global rate is used. In implementation, we perform three rounds of training without expanding tree to tackle cold start problem, since the initial training rounds are sensitive to sampling rate choices, and cap the FedHOO search tree at depth 5. This provides a fine enough resolution of sampling rates while keeping the exponential branching factor computationally manageable. Empirically, we found that deeper trees do not yield meaningful accuracy gains, as the noise inherent in federated training outweighs the benefit of additional granularity, whereas depth 5 provides a stable balance between exploration and efficiency.

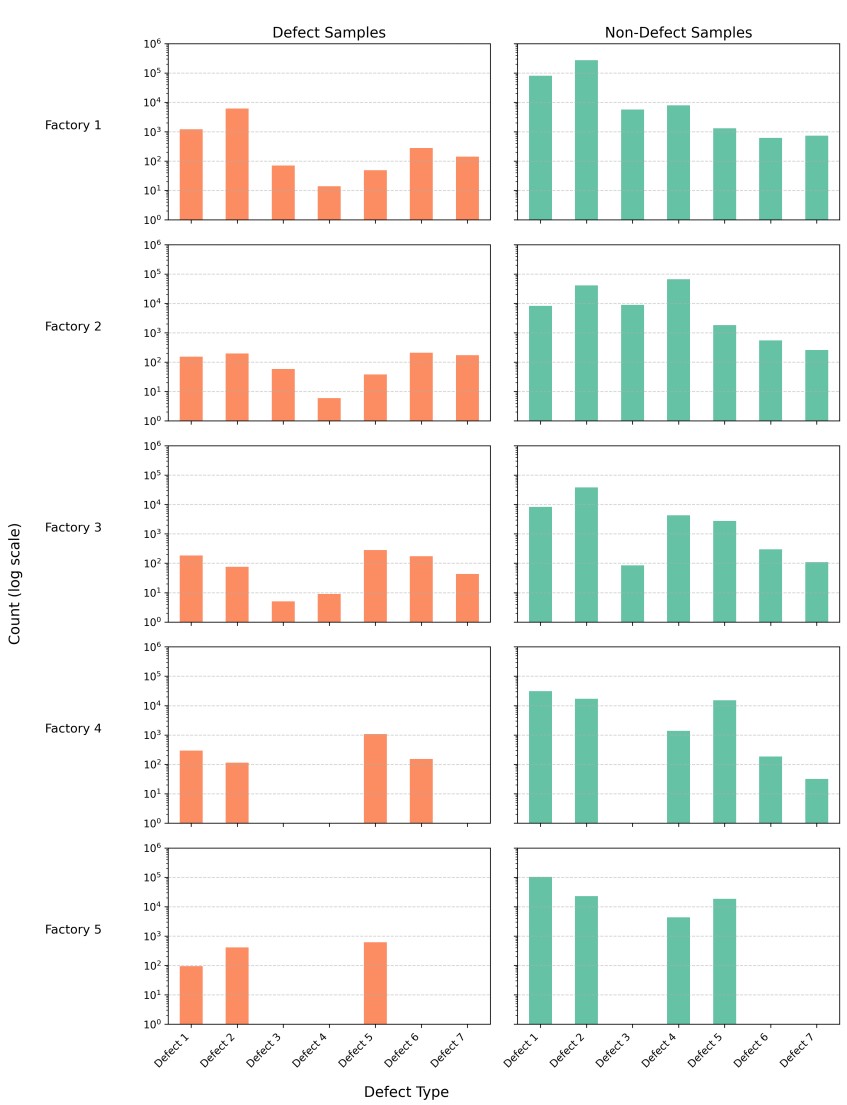

Figure 5: Distribution of sample counts by factory and defect code. Each row represents one factory; the left panel displays defect counts and the right panel shows non-defect counts, both on a logarithmic y-axis.

In the small-scale regime ($K = 5$), all clients participate each round. In the moderate-scale regime ($K = 20$), we sample 50 % of clients per round. Public datasets are split IID across clients; the CDD corpus is partitioned by factory based on real meta data to simulate non-IID conditions.

# D  ADDITIONAL EXPERIMENT RESULTS

In this appendix, we provide supplementary analyses to validate our design choices and assess the proposed algorithms under more challenging conditions. We first verify the optimality of our variance-based grouping threshold, then evaluate the effectiveness of FedHOO, analyze its resampling rate behavior compared to standard HOO, and further test FedCGNM under non-IID data, large-client federations, and varying numbers of groups.

## D.1  VALIDATION OF THE VARIANCE-BASED GROUPING THRESHOLD

To confirm that our grouping rule reliably identifies the optimal split, we sweep the threshold $t \in \{1, \ldots, 9\}$, i.e. the number of rarest classes assigned to the minority group, and record test accuracy on CIFAR-10-LT under imbalance rates 20 and 100. As shown in Figure 6, the threshold $t = 6$ selected by minimizing within-group variance (vertical dashed red line) coincides with the highest performance in both settings, demonstrating that our data-driven rule matches the empirical optimum without exhaustive search.

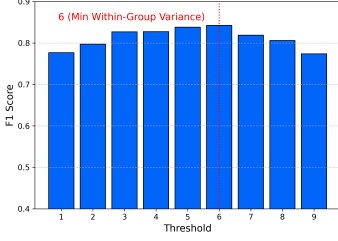

(a) CIFAR-10 - Imbalance rate=20

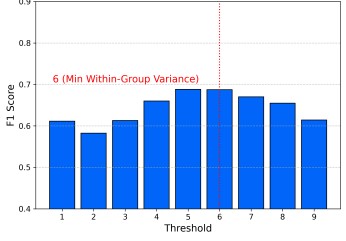

(b) CIFAR-10 - Imbalance rate=100

Figure 6: Test accuracy versus the number of rarest classes assigned to the minority group on CIFAR-10-LT. The dashed red line at $t = 6$ marks the threshold chosen by our variance-based grouping rule, which aligns with the peak test accuracy in both imbalance scenarios.

## D.2  EFFECTIVENESS OF FEDHOO

Table 5 compares the company's metric obtained on the proprietary CDD benchmark when each federated optimizer runs with a fixed uniform resampling rate versus when per-client rates are tuned by our FedHOO strategy. We include this result to demonstrate that FedHOO remains effective even when integrated with alternative training paradigms. FedHOO consistently boosts performance, confirming that our method is beneficial even when combined with optimizers other than FedCGNM.

Table 5: Performance improvement on CDD benchmark with a uniform global sampling rate versus the FedHOO.

| Algorithm | Global Rate | FedHOO | improvement % |
|---|---|---|---|
| FedAvg | 72.83 | 85.37 | 17.22 |
| Weighted CE | 76.58 | 87.20 | 13.86 |
| Ratio Loss | 77.68 | 87.59 | 12.76 |
| FedCGN | 85.37 | 88.25 | 3.37 |
| FedCGNM | 86.88 | 90.87 | 4.59 |

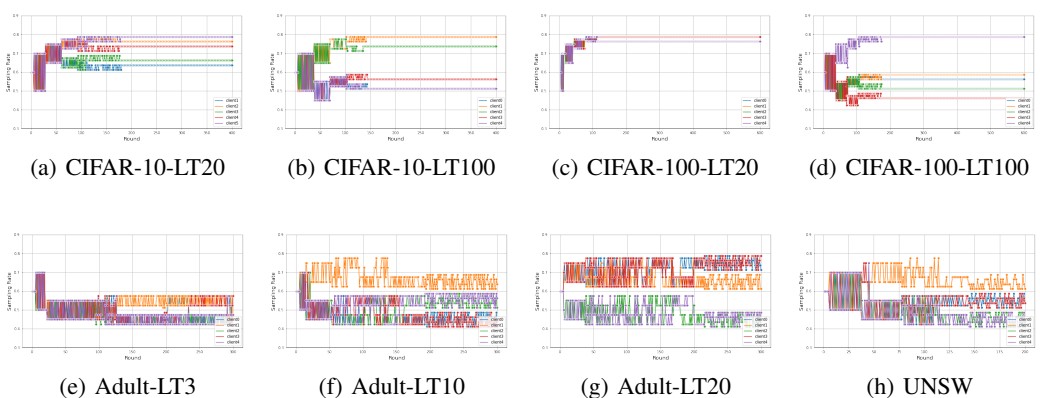

| (a) CIFAR-10-LT20 | (b) CIFAR-10-LT100 | (c) CIFAR-100-LT20 | (d) CIFAR-100-LT100 |
|---|---|---|---|

| (e) Adult-LT3 | (f) Adult-LT10 | (g) Adult-LT20 | (h) UNSW |
|---|---|---|---|

Figure 7: Per-client resampling rate trajectories selected by FedHOO across datasets and imbalance settings.

### D.3 RESAMPLING RATES IN FEDHOO

To better understand and illustrate the benefit of FedHOO, we plot and compare the trajectories of sampling rate selection under FedHOO and standard HOO. In FedHOO, the selected sampling rates correspond to the candidates yielding the best-performing model in each round, which is then adopted as the subsequent training initialization. As outlined in the implementation details, we restrict the search tree to a maximum depth of five.

Figure 7 illustrates how FedHOO adaptively selects per-client resampling rates across different datasets and imbalance severities, and Figure 8 shows how HOO selects per-client resampling rates on CIFAR-10-LT20 dataset. Each curve corresponds to one client, with the $y$-axis showing its selected sampling rate, and the $x$-axis showing the communication rounds.

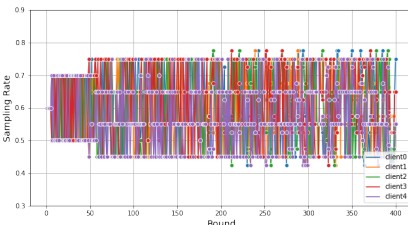

Figure 8: Per-client resampling rate trajectories selected by standard HOO on CIFAR-10 iid setting with imbalance rate $\xi = 20$.

Under standard HOO, the algorithm explores only one branch of the search tree per round. As a result, the sampling rate assigned to each client fluctuates heavily throughout training, and fail to converge in reasonable training time. The per-client trajectories remain noisy even after hundreds of rounds, reflecting the limited feedback that HOO gathers in each step.

In contrast, FedHOO leverages federated parallelism: by evaluating two candidate rates per client and linearly combining their updates, it effectively observes all $2^K$ combinations at once. This parallel exploration dramatically accelerates the search. The sampling rates quickly stabilize after the initial rounds, with each client settling into a distinct but consistent rate. The stabilized patterns observed across datasets in Figure 7 confirm that FedHOO both reduces variance and identifies effective configurations much earlier in training.

Overall, these plots highlight the key difference: while HOO suffers from slow and noisy adaptation due to sequential exploration, FedHOO achieves rapid and stable convergence by exploiting the structure of federated training.

### D.4 FedCGNM in Non-IID and Large-Federation Regimes

To further assess robustness, we evaluate FedCGNM under two challenging conditions beyond the main experiments: (i) non-IID data distributions and (ii) large numbers of clients with partial participation.

**Non-IID distributions.** We simulate client heterogeneity using Dirichlet partitioning with concentration $\alpha = 0.5$. Table 6 shows results on CIFAR-10 and CIFAR-100 with imbalance rate 20. Across both small ($K = 5$) and moderate ($K = 20$) federations, FedCGNM consistently outperforms FedAvg, FedCGN, and other strong baselines, demonstrating its ability to remain effective even when client data distributions are highly skewed.

Table 6: Performance under non-IID distributions (Dirichlet $\alpha = 0.5$).

| Dataset | $K$ | Imbalance | FedAvg | FedCGNM | FedCGN | Weighted CE | Ratio Loss |
|---------|-----|-----------|--------|---------|--------|-------------|------------|
| CIFAR-10 | 5 | 20 | 0.7845 | **0.8316** | 0.8223 | 0.8071 | 0.7952 |
| | 20 | 20 | 0.6942 | **0.7437** | 0.7361 | 0.5616 | 0.6518 |
| CIFAR-100 | 5 | 20 | 0.4150 | **0.4351** | 0.4210 | 0.3511 | 0.4083 |
| | 20 | 20 | 0.3347 | **0.3463** | 0.3390 | 0.2604 | 0.2977 |

**Large-client regime with partial participation.** We also evaluate FedCGNM when the number of clients is large and only small fraction participate per round, which is a scenario that amplifies variance and typically harms optimization. Table 7 reports results on CIFAR-10-LT and CIFAR-100-LT with imbalance rates 20 and 100. While all methods degrade in this setting, FedCGNM consistently achieves the best performance, surpassing both reweighting-based and per-group gradient-normalization baselines.

Table 7: Performance in large-client federations with partial participation.

| Method | C10-LT20 | C10-LT100 | C100-LT20 | C100-LT100 |
|--------|----------|-----------|-----------|------------|
| FedAvg | 0.3324 | 0.2267 | 0.0830 | 0.0471 |
| FedWL | 0.4423 | 0.3131 | 0.0962 | 0.0533 |
| FedRL | 0.3306 | 0.2266 | 0.0847 | 0.0464 |
| FedCGN | 0.4441 | 0.2994 | 0.1226 | 0.0696 |
| FedCGNM | **0.4675** | **0.3331** | **0.1227** | **0.0725** |

Together, these results demonstrate that FedCGNM retains its advantage in more challenging federated environments: it scales to non-IID data and large-client regimes where baseline methods suffer the most.

### D.5 The Effect of the Number of Groups

To confirm that two groups offer the best bias–variance trade-off beyond the setting in Figure 4, we repeat the analysis on other settings. Figure 9 shows the training and validation losses when the number of groups $H$ is set to $\{2, 3, 4\}$ for CIFAR-10 and CIFAR-100.

In both benchmarks the pattern mirrors our earlier finding: validation loss stays lowest and most stable for $H = 2$, whereas $H = 3, 4$ starts to drift upward sooner, which is a signal of accelerated over-fitting. The training loss, by contrast, continues to fall for all values of $H$, which widens the train–validation gap when more than two groups are used. These results reinforce the conclusion that splitting classes into exactly two groups strikes the right balance between reducing gradient variance and controlling over-fitting for CIFAR-10 and CIFAR-100.

### D.6 Per-Class Accuracy Distribution

From Figure 10, FedCGNM (blue) shifts the per-class accuracy distribution markedly to the right compared to FedAvg (orange), substantially reducing the number of near-zero classes. Whereas

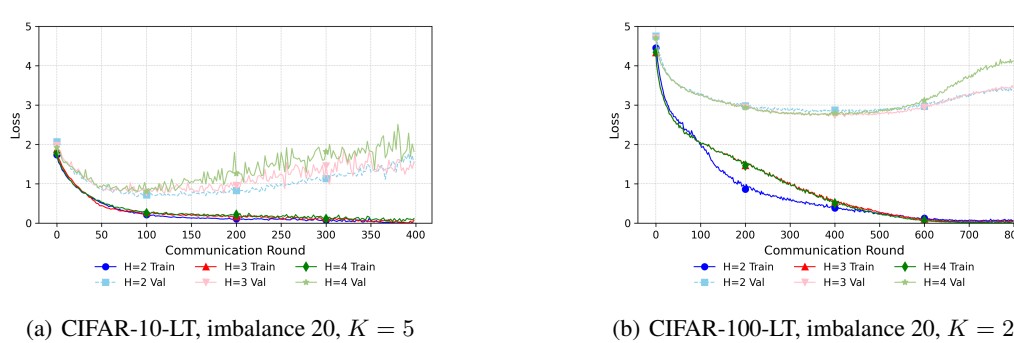

(a) CIFAR-10-LT, imbalance 20, $K = 5$          (b) CIFAR-100-LT, imbalance 20, $K = 20$

Figure 9: Training and validation loss for different numbers of groups $H \in \{2, 3, 4\}$.

FedAvg produces a heavy tail of classes below 0.1 accuracy, FedCGNM elevates most classes into a moderate accuracy range. FedCGN also achieves a more balanced distribution, confirming the effectiveness of the grouping strategy in balancing class-wise performance.

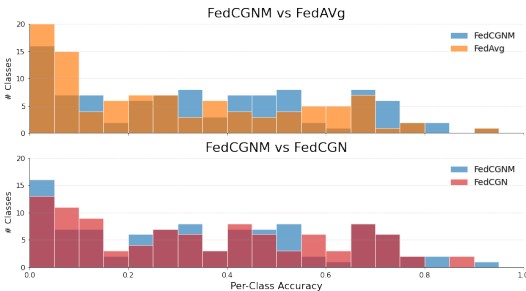

Figure 10: Distribution of per-class test accuracies on CIFAR-100-LT (imbalance rate=100, K=5).

