# OpenReview forum: "Class-Grouped-Normalized-Momentum and Faster Hyperparameter Exploration to Tackle Class Imbalance in Federated Learning"
_ICLR.cc/2026/Conference — ICLR 2026 Conference Withdrawn Submission_

### Official Review · Reviewer_5Aj2 · 2025-10-19

**Soundness:** 3
**Presentation:** 2
**Contribution:** 3
**Rating:** 4
**Confidence:** 3

**Summary:**

This paper addresses the challenge of class imbalance in federated learning by introducing FedCGNM, a client-side optimizer that groups classes by frequency and applies normalized momentum per group to balance majority–minority contributions and reduce gradient noise, together with FedHOO, an X-armed-bandit–based algorithm that efficiently explores local resampling rates by exploiting federated parallelism. The authors provide an convergence analysis, and extensive experiments show improvements over existing baselines.

**Strengths:**

1. The grouped normalized momentum design is simple yet principled and effectively mitigates class imbalance at the optimization level.
2. Theoretical analysis is rigorous and extends normalized-momentum convergence results to the federated context.
3. Empirical evaluations are comprehensive, covering multiple datasets, imbalance ratios, and federation scales, with consistent improvements.

**Weaknesses:**

1. The convergence analysis depends on a strong bounded-gradient assumption (Assumption 3.2), which may not hold in realistic federated environments; many recent works achieve similar guarantees using only the standard bounded-variance assumption, making this assumption unnecessarily restrictive. Moreover, since FedCGNM already normalizes gradients to unit length, this assumption seems redundant, casting doubt on the necessity and validity of the theoretical analysis built upon it.
2. The claim that Per-Class Normalization (PCN) has two limiations and “does not work well even for a moderate number of classes” lacks theoretical or empirical support; the paper neither analyzes the effect of class count on scaling mismatch nor provides experimental evidence of PCN’s degradation. In fact, the PCN-style baseline (FedCGN) performs reasonably well, suggesting that the improvements of FedCGNM mainly stem from added stability rather than fixing a clear failure of PCN.
3. The experimental comparison omits baselines such as FedProx, Scaffold.
4. Some parts of the methodology, particularly the FedHOO description, are notation-heavy and could benefit from more intuitive explanations.
5. It is unclear whether the evaluation considers metrics beyond accuracy or F1; given the focus on imbalanced learning, the inclusion of the AUC metric would provide a more balanced assessment of performance.

**Questions:**

See Weaknesses above.

---

### Official Review · Reviewer_4fW1 · 2025-10-29

**Soundness:** 2
**Presentation:** 3
**Contribution:** 2
**Rating:** 4
**Confidence:** 5

**Summary:**

This paper proposes FedCGNM and FedHOO to address poor minority-class performance caused by global long-tailed imbalance in federated learning. FedCGNM replaces per-class normalization with a small number of groups, which mitigates update scale mismatch and suppresses directional noise; the grouping threshold is chosen by a data-driven “within-group variance minimization” rule. FedHOO formulates the combinatorial hyperparameter search as an X-armed bandit and exploits federated parallelism to evaluate 2^K candidate combinations per round, enabling fast and privacy-friendly hyperparameter exploration in small-K settings.

**Strengths:**

(1) Replacing per-class normalization with “class grouping + normalized momentum” is a simple design and shows some effectiveness in the reported experiments.

(2) With only two local runs per client, the server can evaluate 2^K combinations, which accelerates combinatorial hyperparameter search for small federations.

(3) Broad experimental coverage, including a real industrial use case; the method consistently improves performance on CIFAR-10/100-LT, Adult, UNSW, and a 780k-chip dataset, with the industrial results particularly convincing.

**Weaknesses:**

- Problem setup is unclear. The abstract states “Local class imbalance rooted at global imbalance poses a critical challenge in federated learning,” but local class imbalance may stem from client-side non-IID as well as global imbalance. For example, under global imbalance with IID partitioning, local imbalance still exists, but the challenges differ from those in non-IID settings. The paper should clearly distinguish global imbalance, local imbalance (from global vs. from non-IID), and general data heterogeneity, explain their relationships and distinct challenges, and specify which type is targeted in each experimental setting.

- Per-Class Normalization (PCN) is not a common method in class-imbalance learning and, in my experience, performs much worse than mainstream methods (e.g., LA loss, LDAM-DRW) in centralized training. The paper does not provide a convincing rationale for why a method that underperforms in standard class-imbalance settings would work well in federated learning. Moreover, many works combine these mainstream losses to handle imbalance and non-IID in FL (e.g., [1][2][3]), but there is little discussion or comparison here.

- The paper replaces class-level gradient norms with group-level norms, but appears to lack direct comparisons against per-class gradient normalization. In addition, it does not convincingly explain why H=2 is best. Intuitively, as H increases and approaches per-class normalization, performance might be expected to improve; the paper should clarify why this intuition fails and provide supporting evidence.

- FedHOO lacks convergence/regret analysis. Also, the server linearly combines two local deltas to “synthesize” 2^K global models, which implicitly assumes local updates vary approximately linearly with the sampling rate for small steps. However, non-linearities (momentum, normalization, weight decay, data order) may break this equivalence. Please quantify the gap by comparing synthesized models versus models actually trained under those combinations (e.g., parameter norm differences or validation curves).

- Assumption 3.4 is rather abstract; please provide an operational, testable criterion or empirical evidence to support it.

- The paper reports only F1 and omits standard accuracy on class-balanced test sets. In image tasks, FedCGNM/FedCGN use GroupNorm while baselines use BatchNorm, which may introduce a non-method advantage; please unify normalization across methods or add matched-normalization baselines.

- Code is not released, raising concerns about reproducibility.

[1] No Fear of Classifier Biases: Neural Collapse Inspired Federated Learning with Synthetic and Fixed Classifier

[2] Federated Learning with Label Distribution Skew via Logits Calibration

[3] Aligning model outputs for class imbalanced non iid federated learning

**Questions:**

Refer to Weakness.

---

### Official Review · Reviewer_H9AG · 2025-10-30

**Soundness:** 2
**Presentation:** 3
**Contribution:** 2
**Rating:** 4
**Confidence:** 4

**Summary:**

This paper tackles the problem of class imbalance in federated learning (FL) by introducing two key contributions: FedCGNM (Class-Grouped Normalized Momentum) and FedHOO (Hierarchical Optimistic Optimization). FedCGNM is a client-side optimizer that groups classes (e.g., majority vs. minority) and applies normalized momentum updates within each group to balance gradient contributions and reduce directional noise. In contrast, FedHOO is an X-armed bandit–based algorithm designed to efficiently tune client-specific resampling rates in small-scale federations. The authors argue that prior approaches such as PCN suffer from gradient directional noise and scaling mismatches when dealing with a large number of classes. FedCGNM alleviates these issues by grouping classes, maintaining separate momentum for each group and normalizing them to unit length, while FedHOO leverages the parallelism of FL to explore exponentially many sampling rate combinations at linear computational cost. The paper provides a theoretical convergence analysis for FedCGNM and presents comprehensive experiments on four public long-tailed benchmarks and a proprietary chip-defect classification dataset, demonstrating consistent and robust performance gains over strong baselines.

**Strengths:**

**1.** FedCGNM introduces a novel optimization mechanism to address class imbalance at the gradient level by grouping classes and applying normalized momentum within each group. This approach represents a clear departure from conventional techniques based on per-class normalization, data resampling, or loss-level reweighting.

**2.** The method is extensively evaluated on multiple datasets—CIFAR-10-LT, CIFAR-100-LT, Adult Income, UNSW-NB15, and a real-world chip-defect dataset—under both IID and non-IID settings, consistently demonstrating robust and generalizable performance improvements.

**3.** he paper provides a convergence analysis showing that FedCGNM attains the standard \mathcal{O}(T^{-1/2}) convergence rate under smoothness and bounded variance assumptions, thereby offering strong theoretical support for the proposed method.

**4.** FedHOO effectively leverages federated parallelism to explore client-specific sampling rates in an efficient manner, particularly benefiting small-client federations and addressing a key practical challenge in FL hyperparameter tuning.

**Weaknesses:**

**1.** Although several baseline methods are included (e.g., FedAvg, FedAvg with Weighted CE, FedAvg with Ratio Loss, and FedCGN), more recent federated learning approaches targeting class imbalance—such as FedNoRo and FAST—are only briefly discussed in the related work section. A direct empirical comparison with these methods would strengthen the paper’s claims regarding the superiority of FedCGNM and FedHOO.

**2.** The proposed FedHOO algorithm is evaluated only in small-client settings (K=5), and the authors acknowledge its computational overhead when scaling to larger federations. This limitation constrains its applicability to real-world, large-scale FL deployments.

**3.** The theoretical analysis assumes static resampling rates and fixed groupings, which may not reflect practical dynamics. The impact of adaptive or evolving grouping strategies on convergence behavior remains unexplored.

**4.**  While the variance-based grouping rule is both motivated and empirically supported, it is compared only against a naive half-split baseline. Evaluating other grouping strategies would provide deeper insight into the robustness of this design choice.

**5.** The inclusion of public datasets and detailed appendices—covering grouping thresholds, hyperparameters, and FedHOO pseudocode—enhances reproducibility. However, the use of a proprietary chip-defect dataset limits transparency for part of the evaluation, and the absence of a code release statement slightly weakens the paper’s reproducibility claim.

**Questions:**

**1.** The current experiments primarily evaluate FedCGNM under a two-group configuration (H = 2) and on datasets with a relatively small number of classes. Although Section 4.4 (“Effect of Grouping and Number of Groups”) provides an analysis for H = 2, 3, and 4 on CIFAR-10 and CIFAR-100, this evaluation is insufficient to fully demonstrate the performance of FedCGNM on more complex multi-class datasets with a larger number of categories. It would be valuable to further investigate the method’s scalability and effectiveness on such datasets. Moreover, is there a heuristic or principled approach for selecting the optimal number of groups (H) for a given dataset?

**2.** The convergence analysis assumes static resampling rates and fixed groupings. How does the model’s performance change when these parameters are updated dynamically during training? Are there any observed stability or convergence trade-offs?

**3.** Could the authors provide more details on the computational and communication overhead introduced by FedHOO compared to standard grid search or random search methods, particularly as the number of clients K increases?

**4.** Have the authors considered comparing the variance-based grouping rule with other potential grouping strategies besides the naive half-split method, to verify the robustness and generality of this design choice?

---

### Official Review · Reviewer_Qtuj · 2025-10-31

**Soundness:** 2
**Presentation:** 3
**Contribution:** 1
**Rating:** 2
**Confidence:** 5

**Summary:**

The paper presents two methods used in federated learning. First, they propose FedCGNM a method in which each client optimizer partitions classes into smaller groups and normalizes each group momentum to unit length. By doing this it constructs a gradient that does not penalized any class. Second, the paper introduces a XAB based algorithm that exploits federated parallelism at linear cost. The authors show improvements empirically, and have standard convergence results.

**Strengths:**

The paper studies a relevant problem. I believe class imbalance is a relevant problem that needs to be properly addressed. The tools presented in this paper do respect the constraints of the problem, in that they are private given that they do not share any data from the individual clients.

**Weaknesses:**

The method requires to have at least one sample per class. This is a strong assumption, and one that might not be met in practice. To make matters worse, the authors do no explicitly state this limitation (eqn 6 from algorithm).

The method requires the centralized server to carry out possibly expensive computations when choosing the sampling strategy (line 3 of algorithm). This might no be doable in practice, and has the drawback that requires extra communication steps, and full synchrony.

Regarding the proof, Theorem 3.5 states that FedCGNM convergence does not depend on local computations. Actually, given the square dependance of D_1 and D_2 on E, increasing the local steps is worse for convergence. This is not what has been seen in practice in FL in general. Therefore, this attempt against the use of distributed computing, and therefore the method itself.

**Questions:**

- Why did the authors ignore so many relevant papers that study the same problem they are considering?

CLIMB - https://openreview.net/pdf?id=Xo0lbDt975
https://ieeexplore.ieee.org/abstract/document/9616052
https://ieeexplore.ieee.org/abstract/document/9825928

- Can the authors add relevant baselines to their work? It is not possible to understand the relevance of the work if it is not properly put in place with other existing baselines. See previous point.

- Can the authors provide a citations for the claim:
"However, standard XAB methods (e.g., exhaustive search or HOO) converge too slowly
under FL’s limited rounds and early sensitivity to hyperparameters"

**Details Of Ethics Concerns:**

None.

---

### Note · Authors · 2025-11-19

**Comment:**

We thank the reviewers for their detailed and constructive feedback. Several comments, particularly those regarding the need for additional experimental baselines, are valid, while we disagree with certain points raised about the theoretical analysis.
To ensure that a future version of this work includes a more comprehensive empirical study, we have decided to withdraw the submission at this time. We appreciate the reviewers’ efforts and will incorporate the helpful suggestions into a revised manuscript.

**Withdrawal Confirmation:**

I have read and agree with the venue's withdrawal policy on behalf of myself and my co-authors.